# The Role of the Immune System in IBD-Associated Colorectal Cancer: From Pro to Anti-Tumorigenic Mechanisms

**DOI:** 10.3390/ijms222312739

**Published:** 2021-11-25

**Authors:** Sofía Frigerio, Dalia A. Lartey, Geert R. D’Haens, Joep Grootjans

**Affiliations:** 1Department of Gastroenterology and Hepatology, Amsterdam University Medical Centers, Location AMC, 1105 AZ Amsterdam, The Netherlands; s.n.frigerio@amsterdamumc.nl (S.F.); d.a.lartey@amsterdamumc.nl (D.A.L.); g.dhaens@amsterdamumc.nl (G.R.D.); 2Tytgat Institute for Liver and Intestinal Research, Amsterdam University Medical Centers, Location AMC, 1105 AZ Amsterdam, The Netherlands

**Keywords:** IBD, IBD-associated cancer, inflammation, immune system, immune cells, immunosurveillance, immunosuppression, colitis-associated cancer, colorectal cancer

## Abstract

Patients with inflammatory bowel disease (IBD) have increased incidence of colorectal cancer (CRC). IBD-associated cancer follows a well-characterized sequence of intestinal epithelial changes, in which genetic mutations and molecular aberrations play a key role. IBD-associated cancer develops against a background of chronic inflammation and pro-inflammatory immune cells, and their products contribute to cancer development and progression. In recent years, the effect of the immunosuppressive microenvironment in cancer development and progression has gained more attention, mainly because of the unprecedented anti-tumor effects of immune checkpoint inhibitors in selected groups of patients. Even though IBD-associated cancer develops in the background of chronic inflammation which is associated with activation of endogenous anti-inflammatory or suppressive mechanisms, the potential role of an immunosuppressive microenvironment in these cancers is largely unknown. In this review, we outline the role of the immune system in promoting cancer development in chronic inflammatory diseases such as IBD, with a specific focus on the anti-inflammatory mechanisms and suppressive immune cells that may play a role in IBD-associated tumorigenesis.

## 1. Introduction

Ulcerative Colitis (UC) and Crohn’s Disease (CD) are the two forms of inflammatory bowel disease (IBD), characterized by chronic inflammation of the digestive tract leading to diarrhea, rectal bleeding and abdominal pain. One of the most severe complications of IBD is the development of colorectal cancer (CRC). It is generally accepted that the continuous exposure of intestinal epithelial cells (IECs) to proinflammatory stimuli and excessive cell damage with increased IECs turnover results in both genetic and immunological alterations, making IBD patients prone to developing CRC [1,2,3]. Population-based evaluations estimate that UC increases the risk of developing CRC two to three-fold compared to the non-IBD population [4,5] and patients with CD have also been reported to have increased CRC risk when compared to the general population [6,7]. The risk of developing CRC increases significantly after 10 years of IBD diagnosis and is higher in patients with continuous chronic intestinal inflammation, highlighting the crucial role of a prolonged inflamed environment in the pathogenesis of IBD-associated cancer [8,9,10]. Compared to sporadic CRC, IBD-associated cancer has worse overall survival rates. The reported differences in clinical features, disease pathogenesis and epidemiologic characteristics, which may be explained by differences in tumor biology, have recently been excellently reviewed [11].

IBD-associated tumorigenesis follows a multistep sequence of genetic and morphological alterations in the intestinal epithelium, from inflamed intestinal mucosa to dysplasia and finally carcinoma [12,13]. The genetic mutations that drive the progression from dysplasia to carcinoma in IBD-associated cancer have been extensively studied and include mutations in tumor suppressor genes (TP53 mutations) and in genes that regulate the cell cycle and cell proliferation such as KRAS and APC mutations (Figure 1). Surprisingly, there is very little difference in the genetic mutations that drive carcinogenesis among IBD-associated cancer and sporadic CRC [14,15].

As IBD-associated dysplasia develops in the background of chronic inflammation, studies have highlighted the role of pro-inflammatory cytokines on tumorigenesis. Most observations were made in mouse models [16,17,18,19,20]. Little is known about the potential role of endogenous anti-inflammatory or immunosuppressive mechanisms that evolve as a consequence of chronic inflammation in the gastrointestinal tract, although such immunosuppressive mechanisms may contribute to tumorigenesis by impairing anti-tumor immunity.

This review will delineate the role of the immune system in IBD-associated cancer with a particular focus on the role of immunosuppressive mechanisms that may be involved in the development of CRC in IBD patients.

## 2. IBD-Associated Cancer: From Inflamed Tissue to Carcinoma

IBD patients are at higher risk of developing CRC, with disease extent and duration being two of the most prominent risk factors. For example, the risk of CRC in UC patients is 1, 2 and 18% after 10, 20 and 30 years of diagnosis, respectively [9,21,22]. However, recent evidence suggests that this risk is decreasing. In fact, an Australian cohort study performed in 2014 showed a cumulative risk of 1% at 10 years, 2% at 20 years and 3% at 30 years post diagnosis [23]. Patients with CD also appear to have an increased risk of CRC, similar to UC patients, yet this is only the case in those CD patients having colonic inflammation, and not in CD patients in which inflammation is confined to the small intestine [24]. This again emphasizes the critical role of chronic inflammation in the pathogenesis of IBD-associated cancer. Another very important risk factor for CRC in IBD patients is concomitant primary sclerosing cholangitis (PSC), which increases the CRC risk by at least 4-fold when compared to UC patients without PSC [25]. The elevated risk of cancer in patients with IBD is also the consequence of the so-called field cancerization effect. In this phenomenon, clonally derived, neoplastic mutant cells form an indistinguishable “field” within the inflamed intestinal segment, meaning that the whole intestinal area is at risk of developing cancer [26].

IBD-associated cancer develops from dysplastic lesions within the colonic mucosa. Colorectal dysplasia can be defined as a neoplastic alteration of the intestinal epithelium that remains confined within the basal membrane [27]. Dysplastic lesions show specific morphological features, such as nuclear alterations, cytoplasmic abnormalities and an abnormal architectural pattern due to uncontrolled cell proliferation [28,29]. In IBD-associated cancer, there is a multi-step sequence of histological and morphological changes, namely from inflamed mucosa to low-grade dysplasia (LGD), high-grade dysplasia (HGD) and eventually carcinoma (Figure 1) [30]. This sequence is different from that in sporadic CRC, where an early adenoma-like lesion emerges from the normal mucosa, after which it evolves to late adenoma and finally adenocarcinoma [31].

Regarding the clinical aspects, CRC associated with IBD is often diagnosed at a more advanced stage compared to sporadic CRC. This is probably due to the difficulty in identifying dysplastic lesions in IBD-associated cancer, as they are usually flat and endoscopically unresectable [1,9,26,32]. The predominance of flat and unresectable lesions is one of the contributing factors to the increased mortality that this cancer presents, with a 1.7-fold increase in mortality among IBD-associated cancer patients in contrast with patients with non IBD and CRC [33]. After detection of LGD in IBD patients, guidelines recommend an intensified surveillance program or even a total colectomy due to high risk of developing cancer [30,33]. As an example, a recent study in a large IBD cohort with a history of LGD found a cumulative incidence of advanced neoplasia of 22% up to 15 years after LGD detection [34]. Moreover, around 25% of all HGD and carcinoma lesions were detected within the first year after LGD detection [34], further emphasizing the risk of cancer progression in patients with IBD-associated dysplasia.

Given the unquestionable role of intestinal inflammation in IBD-associated cancer development, an optimal control of inflammation in IBD patients should prevent cancer development. The use of anti-inflammatory drugs that induce mucosal healing, together with appropriate endoscopic surveillance is considered the optimal strategy to prevent cancer development [35,36,37].

Although there is limited evidence of the direct protective effect of most of the biological therapies on the development of inflammation-induced cancer, a recent meta-analysis demonstrated that the use of 5-aminosalisylate (5-ASA) and thiopurines are protective factors for IBD-associated advanced neoplasia [33].

5-ASA is the most extensively used drug as maintenance therapy in UC [38,39]. Interestingly, mouse studies demonstrated that this drug reduced tumor growth in CAC mouse models, but not in sporadic CRC mouse models [40]. 5-ASA reduces tumor growth by affecting multiple molecular pathways. For instance, it reduces beta-catenin accumulation in APC-mutated cells and improves replication fidelity by reducing the occurrence of DNA mutations [41,42]. Meta-analyses regarding the protective effect of 5-ASA on CRC development showed conflicting results, probably due to heterogeneity in the populations studied. Moreover, it remains unclear whether it has an intrinsic protective effect, or whether protection is related to improved mucosal healing [43,44].

Thiopurines have also been suggested as chemo-preventive agents in IBD-associated cancer [45,46], although studies have shown conflicting results, likely again due to high heterogeneity in the studied cohorts and potential confounding factors. For example, no protection against CAC was observed with the use of thiopurines in the French CESAME cohort [47]. In contrast, a recent meta-analysis demonstrated a protective role of thiopurines in CAC prevention among UC patients with HGD [48].

Taken together, IBD-associated cancer represents a relevant clinical problem, which is reflected in its high mortality rate and the difficulty of early detection, diagnosis and treatment. In order to improve clinical management and to better predict which patients are more likely to develop carcinomas from dysplastic lesions, it is critical that we understand the pathophysiology of IBD-associated dysplasia and cancer.

## 3. The Role of the Immune System in Cancers Developing on the Background of Chronic Inflammation

Chronic inflammation results in continuous tissue damage, accumulation of immune cells and fibrosis. This dysregulated and chronically inflamed environment can be observed in the setting of many autoimmune disorders with recurrent episodes of acute inflammation, and in patients with long-standing chronic infections [49]. It is well established that such a chronically inflamed environment is associated with the development of cancer, as approximately 25% of cancers develop in the background of such chronic inflammation and/or chronic infection [50,51,52]. For example, *Helicobacter pylori* gastritis is associated with gastric cancer [53,54,55], human papillomavirus infection with cervical cancers [56,57], and chronic hepatitis B and C infection is associated with hepatocellular carcinoma [58,59]. In addition to microbial-induced inflammation, inflammatory auto-immune conditions, including IBD and PSC, are associated with the development of CRC and cholangiocarcinoma, respectively [9,11,13,21,60,61].

The current consensus is that chronic inflammatory conditions lead to genetic mutations, genomic instability and DNA damage, as well as to immune cell dysregulation, which collectively promotes cancer development [1,3,62,63,64]. Indeed, many pro-inflammatory mediators that are present in a chronically inflamed environment can drive neoplastic transformation, such as cytokines interleukin (IL) -1β, IL-6, Tumor Necrosis Factor alpha (TNF-α) [65,66,67,68], other molecules and transcription factors (prostaglandin E2, S100A8/9 proteins, STAT3-mediated signaling) [69,70,71,72,73,74,75,76,77], as well as chemokines: C-C motif ligand (CCL) -2, CCL5, CCL22, C-X-C motif ligand (CXCL) -5, CXCL12 [78,79,80]. In particular, CXCL12 secreted by stromal fibroblasts can bind to its receptor in tumor cells and stimulate motility and chemotaxis [81]. In addition, it has been demonstrated that neutralizing antibodies against CCL2 and CXCL8 prevent the formation of lung metastases and inhibit tumor growth in breast carcinoma and prostatic cancer mouse models, respectively, demonstrating their pro-tumorigenic role [82,83].

However, apart from pro-inflammatory mediators, endogenous anti-inflammatory mechanisms that evolve during chronic inflammation may also shape an environment in which cancer cells thrive (Figure 2). Unbalanced immune and non-immune secreted compounds may result in a shift to an immunosuppressive microenvironment, affecting the immune system’s ability to eliminate neoplastic cells. Thus, a chronic pro-inflammatory environment may not only be associated with cancer formation through continuous tissue damage and presence of pro-inflammatory tumor-driving mediators, but also through a chronic inflammation-induced immunosuppressive state, which interferes with the generation of effective anti-tumor immune responses.

First, tissue-infiltrating monocytes and tissue-resident macrophages display a shift towards an anti-inflammatory M2 phenotype under chronic inflammatory conditions [84,85]. These cells elicit several suppressive functions, including impairment of effector activity of T cells and dendritic cells (DCs) by anti-inflammatory cytokine production [86,87,88]. In addition, M2-like macrophages stimulate cell proliferation via epidermal growth factor production and angiogenesis, which facilitates tumor invasion [89,90,91]. In addition, chronic inflammation leads to the recruitment of myeloid-derived suppressor cells (MDSCs) (Figure 2), immature immune cells specialized in suppressing T cell effector functions and proliferation [92]. MDSCs also produce anti-inflammatory cytokines and reduce expression of activation markers on natural killer (NK) cells [93], all of which lead to a defect in immunosurveillance and promote cancer development.

Second, mounting proper immune responses against dysplastic/tumor cells requires recognition of tumoral antigens by DCs, which then prime naive T cells in secondary lymphoid organs. In chronically inflamed tissues, however, this crucial DC function may be impaired as a consequence of an increase in tolerogenic DCs with a low capacity of antigen presentation and decreased class-II major histocompatibility complex expression [94], which is associated with defective antigen presentation to naive T cells and thus impaired T cell anti-tumor immunity. This is critical, as T cells are the most important orchestrators of the anti-cancer immune response, both through direct tumoral cell killing by cytotoxic T CD8^+^ cells, and through activation of anti-tumor immune responses with the help of T CD4^+^ helper cells type 1 (Th1) [95].

Next to this impaired function of T cells, chronic inflammation leads to regulatory T cells (Tregs) recruitment [96] (Figure 2). Tregs are immunosuppressive in that they impair the function of T, NK and DC cells by producing anti-inflammatory cytokines (IL-10, transforming growth factor beta (TGF-β)). In addition, they express inhibitory surface markers that facilitate immune-cell suppression, such as programmed cell death protein 1 (PD-1) and cytotoxic T-lymphocyte-associated protein 4 (CTLA-4), among other suppressor mechanisms [97,98,99]. These inhibitory markers can also be induced by pro-inflammatory cytokines and mediators that can be found in a chronic inflamed milieu. For example, interferon-gamma (IFN-γ) can stimulate expression of programmed death-ligand 1 (PD-L1) in lung and colon cancer cell lines [100,101]. Moreover, chronic inflammatory conditions induce PD-1 and the inhibitory marker CTLA-4 in CD4^+^, CD8^+^ T cells and Treg cells in chronic infections due to continuous antigen stimulation [102,103], and negatively regulate T cell activation [104]. Although expression of these inhibitory cell surface proteins is likely involved in controlling excessive inflammation, their possible role in promoting carcinogenesis in a chronically inflamed tissue is unclear.

In summary, chronic inflammatory diseases predispose individuals to an increased risk of cancer by chronic tissue damage, DNA damage and several pro-inflammatory cytokines, but likely also by generating a tumor-promoting anti-inflammatory and suppressive microenvironment.

## 4. The Role of the Immune System in IBD-Associated Cancer

### 4.1. Immune Signaling Pathways in IBD: Contribution to Cancer Onset

#### 4.1.1. NF-κB/TNF-α

NF-κB is a family of transcription factors which play a role in inflammation, cell proliferation and malignant transformation [105]. In both canonical and non-canonical pathways, NF-κB forms a complex with its inhibitor IκB in the cytoplasm. In response to diverse stimuli such as inflammatory cytokines, growth factors or microbial components, IκB is degraded by IκB kinase (IKK complex) and NF-κB is translocated into the cell nucleus as part of the canonical pathway [106]. Once in the nucleus, it regulates the transcription of pro-inflammatory cytokines, chemokines and other inflammatory mediators that will promote and sustain the inflammatory reaction. As a pivotal regulator of the inflammatory responses, NF-κB has been implicated in the pathogenesis of IBD [107]. Furthermore, it has been shown that NF-κB contributes to IBD-associated tumorigenesis, mainly by activating the transcription of pro-inflammatory cytokines and by promoting tumor growth and metastasis via promotion of angiogenesis-related genes and anti-apoptotic genes [108,109] (Figure 3). The crucial role of NF-κB in IBD-associated cancer development has been demonstrated in vivo in a mouse colitis-associated cancer (CAC) model; Azoxymethane (AOM)/dextran sulfate sodium (DSS) model, in which inactivation of the canonical NF-kB signaling pathway through specific ablation of NF-κB kinase B (IKKβ) in intestinal epithelial cells reduced tumor incidence. Interestingly, deletion of the gene encoding IKKβ in myeloid cells decreased both tumor number and size, demonstrating the importance of NF-κB signaling in immune cells in CAC development [110]. In another study in AOM/DSS-treated mice, deletion of NLRP12, a member of the Nod-like receptor family which negatively regulates NF-κB, resulted in severe colitis and increased susceptibility to CAC as compared to wild-type mice [111]. This further demonstrates the important pro-tumorigenic role of NF-κB in IBD and IBD-associated cancer by limiting apoptosis and by promoting the production of pro-inflammatory cytokines.

Moreover, NF-κB is a transcription factor for many pro-inflammatory cytokines, such as TNF-α, a pleiotropic cytokine that promotes IECs apoptosis and acts on innate immune cells via activation of the NF-κB transcription pathway, among other important functions [112,113]. TNF-α seems to play a controversial role in cancer progression in IBD. It can induce epigenetic changes and can promote oncogene expression levels [114]. The pro-tumorigenic role of TNF-α in the progression to cancer under chronic inflammatory conditions was demonstrated in CAC mouse models, in which treatment with anti-TNF-α antibodies decreased both inflammation and the number of intestinal carcinomas [16]. Furthermore, administration of a transmembrane TNF-α antibody in an AOM/DSS mouse model dampened the inflammatory response and tumor formation. Anti-tumorigenic effects were observed both during the inflammation induction cycles with DSS and after CAC development, highlighting transmembrane TNF-α as a candidate target for treatment [115]. Unfortunately, there are currently no human data available that demonstrate a clear effect of anti-TNF antibodies in protection from IBD-associated cancer [116]. However, two cohort studies with long-term follow-up of IBD patients treated with the anti-TNF-α antibody infliximab, did not show increased CRC incidence [117,118].

#### 4.1.2. IL-6/STAT3 and IL-22/STAT3

IL-6 is produced by innate immune cells residing in the intestinal mucosa and regulates survival and proliferation of IECs [119]. Importantly, IL-6 has been demonstrated to have a strong impact on the early progression to carcinoma in IBD via STAT3 signaling (Figure 3). Increased expression of IL-6 and STAT3 was demonstrated in biopsies from UC patients with active disease and in IBD-associated carcinomas, as compared to patients with inactive disease or controls [120]. In CAC mouse models, IL-6 ablation reduced tumor formation and mice injected with recombinant IL-6 had increased tumor load. This was STAT3-dependent, as deletion of STAT3 specifically in IECs resulted in lower tumor numbers and size, as well as a reduced percentage of proliferating cells in colonic crypts [121]. Further research into the possible mechanism of IL-6-mediated tumor promotion showed that macrophage-derived IL-6 attached to a soluble IL-6R is crucial in IL-6 trans-signaling in intestinal epithelial cells and thus in the development of CAC [20]. Thus, several studies have shown that IL-6, and downstream STAT3 signaling, is a critical tumor-promoting cytokine in CAC.

STAT3 may also be indirectly involved in CAC via activation of IL-22 production, for example by CD4^+^ T cells [122]. IL-22 is an important cytokine for epithelial cell and mucus layer regeneration, as well as for the production of antimicrobial compounds and mucosal wound healing [123,124] (Figure 3). However, despite its role in intestinal homeostasis, IL-22 may also possess pro-tumorigenic effects, as increased expression of IL-22, IL-22R and phosphorylated STAT3 have been demonstrated in inflamed and dysplastic intestinal tissues from UC patients [125,126].

#### 4.1.3. NFAT

NFAT is a family of transcription factors mainly expressed in T cells, which regulate important activating signaling pathways. NFAT has been shown to be involved in both IBD pathogenesis and CRC development [127,128,129] and has therefore also been studied in CAC development. In a CAC mouse model, NFATc2-deficiency resulted in decreased tumorigenesis, which was accompanied by reduced intestinal inflammation and decreased production of the pro-inflammatory cytokines IL-6 and IL-21 by CD4^+^ T cells in spleen and lamina propria. Furthermore, the administration of hyper IL-6 abrogated the effect of protection in NFATc2-KO mice [130]. This shows that NFATc2 promotes tumorigenesis in the context of colitis, in a process dependent on IL-6. Apart from NFATc-2, NFATc-3 has been shown to be upregulated in AOM/DSS-treated CAC mice as compared to control mice. Mice deficient in NFATc3 showed a decrease in tumor numbers and size. In addition, NFATc3 inhibited proliferation and epithelial to mesenchymal transition in vitro in a CRC cell line [131]. In conclusion, NFATc2 and NFATc3 are key regulators of intestinal inflammation and are important in the initiation and progression of IBD-associated cancer.

#### 4.1.4. IL23/STAT3/Th17 Signaling Pathway

IL-23 is produced by a variety of immune cells, including antigen presenting cells (APCs), and plays an important role in sustaining the inflammatory response in IBD [132,133]. The IL-23 heterodimer is made up of the p19 and the p40 subunit. Once IL-23 binds to its receptor IL-23R, it activates janus kinases, which will then phosphorylate the receptor, inducing the recruitment of several STAT proteins, including STAT3. Ultimately, IL-23 signaling activates the transcription of effector cytokines genes belonging to the Th17 subtype, such as IL-17A, IL-17F, IL-22 and IFN-γ [134] (Figure 3). The role of the IL23/Th17 pathway in CAC pathogenesis is not completely clear, but IL-17A appears to drive tumor formation in mouse models for inflammation-induced cancer, as both a blockade of IL-17A by anti-IL-17A antibodies [135], and genetic deletion of IL-17A [136] was associated with decreased tumor size and number.

According to the pro-tumorigenic role of the IL-23/Th17 signaling pathway, it could be envisioned that treating IBD patients with anti-IL-23 or anti-IL-17A antibodies would also contribute to a decrease in CAC incidence, both indirectly by decreasing inflammation, but potentially also directly. Ustekinumab, a humanized monoclonal antibody that binds to the p40 subunit, is currently used for the treatment of UC [137]. Even though clinical data are still lacking, preliminary results in a mouse preclinical model identified IL-23 as a promising target to prevent CRC associated with chronic inflammation in IBD [138]. One of the important effects of the use of ustekinumab that needs to be considered as a risk factor for CRC is that blocking p40 is also associated with decreased levels of anti-tumorigenic IL-12 [139,140]. Fortunately, randomized clinical trials reported very low incidence of colorectal cancer patients receiving Ustekinumab as compared to patients receiving placebo, even though the follow-up period was short (52 weeks) [141], and long-term safety data for its use in UC patients will follow.

### 4.2. Immune Cells in IBD and Its Contribution to IBD-Associated Cancer

In this section, we will comment on the role of immune cells in immunosurveillance against cancer and IBD-associated cancer, and on the pro- and anti-inflammatory mechanisms that these cells possess to promote cancer development in the chronic inflamed IBD environment.

#### 4.2.1. Macrophages

Macrophages, mainly tissue-resident cells belonging to the innate immune system, are involved in both anti-inflammatory (M2 or alternatively activated) as well as pro-inflammatory (M1 or classically activated) immune responses. In the intestine, monocytes change into tolerogenic lamina propria macrophages, which are hyporesponsive to microbial stimuli and do not initiate inflammatory responses [142]. In addition, these macrophages produce large amounts of anti-inflammatory cytokines, such as IL-10 and TGF-β, creating a tolerogenic environment and promoting the expansion of CD4^+^CD25^+^ Treg T cells [143]. Biopsies from IBD patients demonstrate a high abundance of monocyte-derived macrophages infiltrating the intestinal mucosa, predominantly of the M1 or pro-inflammatory phenotype [144,145,146]. More recently, it was shown that the non-dysplastic mucosa from IBD patients is populated by both M1 and M2 macrophages [147].

Classically, M1 macrophages participate in anti-tumor immunosurveillance by directly acting as APCs to activate T cells. In addition, macrophages produce cytokines that determine polarization to different effector T-cell subtypes and phagocytose neoplastic cells that are covered with antibodies and other opsonins [148]. In IBD, macrophages have also been proposed to actively play a role in tumorigenesis. M1 proinflammatory macrophages foster a pro tumorigenic environment by producing tumor-promoting cytokines, which exert a well-known proliferative effect on colonic cells via induction of transcription factor NF-κB and signal transducer STAT3 [76,149]. Macrophages have also been identified in IBD-associated dysplastic lesions and the level of macrophage infiltration correlated positively with the number of dysplastic lesions [150]. In line with these results, Khan and colleagues also identified a progressive increase in macrophage density from colitis to dysplasia and cancer [151]. Although the phenotype of these macrophages was not identified, another study showed that the percentage of M2-like macrophages markedly increased in IBD-associated cancers, suggesting a polarization to anti-inflammatory or immunosuppressive macrophages in the transition from dysplasia to carcinoma [152]. In support of this, Kvorjak et al. showed that CD163^+^ M2-like macrophages are significantly increased in intestinal tissue from UC- and IBD-associated cancer patients [153]. Moreover, the authors postulate that CCL17 and IL-13 produced by these CD163^+^ M2 macrophages induce activation of oncogenic pathways involving AKT and STAT6, as well as expression of aberrant glycans on colonic epithelial cells which inhibit activity of DCs and NK cells, thus dampening immunosurveillance (Table 1). Another intriguing anti-inflammatory mechanism by which macrophages may promote tumorigenesis is by the abolishment of T cell proliferation through nitric oxide production. More specifically, it was shown that macrophages that express inducible nitric oxide synthase (iNOS) were capable of suppressing antigen-specific T-cell responses against *Listeria monocytogenes* in the spleen [154] (Table 1). As iNOS+ macrophages are abundant in inflamed intestinal mucosa in IBD, it seems plausible that they also exert a suppressive effect in T cells, affecting immunosurveillance against the malignant transformed cells that appear in the pro-tumorigenic, inflamed milieu. It has also been proposed that macrophages may have a dichotomous effect in IBD-associated cancer via TGF-β activation by the nuclear receptor PPAR-γ. In particular, linoleic acid, a chemical compound present in the diet, activates PPAR-γ, which stimulates macrophages to produce TGF-β, ameliorating colitis symptoms but increasing tumorigenesis in a CAC mouse model [155].

In conclusion, macrophages display both pro- and anti-inflammatory mechanisms that contribute to IBD-associated cancer. It could be envisioned that M1 macrophages initially invade IBD-inflamed mucosa and are involved in carcinogenesis by the production of pro-inflammatory mediators. M1-like macrophages acquire a more M2-like immunosuppressive phenotype as inflammation becomes chronic, and indeed a higher abundance of M2 macrophages with anti-inflammatory properties is observed in dysplastic lesions and carcinomas in IBD (Figure 1). This could further accelerate carcinogenesis as their immunosuppressive phenotype hinders adequate immunosurveillance. To better characterize these mechanisms, more studies into the role of macrophages in IBD-associated cancer are necessary.

#### 4.2.2. Neutrophils and Eosinophils

Apart from monocyte-derived macrophages, other polymorphonuclear cells such as neutrophils and eosinophils are recruited into the inflamed mucosa in IBD [165,166], but their role in promoting IBD-associated cancer has not yet been well addressed. Interestingly, expression of the transcription factor BATF3 in intestinal epithelial cells promoted the transcription of CXCL5, which together with CXC chemokine receptor (CXCR)- 2 stimulate neutrophil recruitment, which was associated with increased CAC development in AOM/DSS-treated mice [167]. Neutrophils have a prominent place in IBD pathophysiology, being a central effector cell in inducing mucosal damage, producing reactive oxygen species, reactive nitrogen species and other specific enzymes that disrupt intestinal tissue [168]. This can also promote dysplasia and carcinoma development in the context of IBD, as neutrophil-derived products generate DNA damage and genetic mutations and result in increased epithelial cell proliferation [169]. Conversely, some studies show that neutrophils contribute to immunosurveillance, and have a protective role in the development of IBD-associated cancer. For example, Zhou et al. reported the presence of a subpopulation of neutrophils, namely CD177^+^ neutrophils, that have tumor-suppressor properties. These neutrophils suppressed tumorigenesis and epithelial proliferation demonstrated by increased expression of the proliferation marker Ki67 in CD177^−/−^ mice. In line with this, patients with IBD-associated cancer and CRC patients with high CD177^+^ neutrophil infiltration had better overall survival compared to the control population [170]. Another study recently showed that neutrophils slow tumor growth by restricting tumor-associated microbiota and IL-17-dependent tumor-associated inflammatory responses [171]. Thus, neutrophils have a dual role in IBD-associated carcinogenesis, being both pro-tumorigenic by generating pro-inflammatory mediators that increase DNA damage and tissue turnover, and anti-tumorigenic by playing an important role in immunosurveillance.

Together with neutrophil infiltration, eosinophils are also commonly present in the inflamed mucosa of IBD patients [172,173]. Activated eosinophils accumulate in the gut of IBD patients and directly contribute to the onset of the inflammatory process in IBD by degranulation [174]. In general, the presence of eosinophils in tumors correlates with a better disease outcome [175,176], and the presence of activated eosinophils has been demonstrated to reduce tumor growth in CRC [177,178]. In IBD-associated cancer, eosinophils seem to play a protective role in tumor development mainly via the effect of the cytokine IL-33. Specifically, IL-33 has been demonstrated to sustain eosinophils by increasing their viability and cytotoxic potentials, making them more effective at fighting malignant cells [179]. However, additional studies are necessary to unravel the function of eosinophils in IBD-associated cancer.

#### 4.2.3. Dendritic Cells

DCs are a heterogeneous group of APCs specialized in the recognition and processing of antigens, and in the activation of naive T lymphocytes. In homeostasis, DCs promote and maintain a tolerogenic immune response. Conversely, in IBD, DCs lose their tolerogenic phenotype and display pro-inflammatory features, causing excessive T-cell responses with the subsequent overload of pro-inflammatory cytokines [180,181,182,183]. This pro-inflammatory milieu may drive cancer formation, yet the subsequent T cell responses also play a key role in cancer immunosurveillance.

Only few studies exist that highlight a specific anti-tumoral function of DCs in IBD-associated cancer. In a mouse model for CAC, IRF8-expressing cDC1s (conventional DCs specialized in activating T CD8^+^ cells) do not seem to contribute to anti-cancer immune responses [184]. Furthermore, it was shown that alterations in phenotype and function of DCs in IBD could play a role in the onset of IBD-associated cancer. First, it was demonstrated that DCs foster a pro-neoplastic inflammatory environment through expression of the transcription factor T-bet [185]. Additionally, it was demonstrated that DCs may exert anti-inflammatory or immunosuppressive mechanisms in the non-dysplastic mucosa from IBD patients, dampening immunosurveillance and contributing to carcinoma progression. For example, it was recently shown that malignant transformation of dysplastic lesions was accelerated in CAC mice in which the transcription factor Notch 2 was deleted in DCs. The authors further demonstrated that Notch2-deficient DCs displayed differentiation defects and had less CC chemokine receptor (CCR)-7 expression, which impeded the migration to the mesenteric lymph nodes and abolished adequate antigen cross-presentation to CD8^+^ T cells (Table 1). This may therefore impact immunosurveillance and favor the dysplasia to carcinoma progression in mice [156]. Interestingly, pDCs could also play a role in the development of IBD-associated cancer through the recruitment of MDSCs, as observed in a mouse model for CAC [157] (Table 1). These MDSCs are known for their suppressor capacity, are increased in several malignancies and have been implicated in tumorigenesis [186,187]. Thus, increased pDCs in IBD are involved in shaping a pro-tumorigenic milieu by recruitment of immunosuppressive cells, including MDSCs. CXCR2-expressing MDSCs are also recruited into the colonic mucosa of AOM/DSS-treated mice, which resulted in inhibition of CD8^+^ T cell effector functions, thereby accelerating tumor growth [158].

Taken together, DCs in IBD-associated cancer may have an anti-inflammatory role, associated with suppressed immunosurveillance. However, it is noteworthy that most of our knowledge regarding the role of DCs in IBD-associated cancer relies on mouse models. Therefore, more insight into the role of DCs in inflammation-associated colorectal carcinogenesis in humans is necessary.

#### 4.2.4. T Cells

The lamina propria intestinal T-cell compartment is mainly composed of effector memory CD4^+^ resident T cells [188]. They are categorized as CD4^+^ helper T cells (Th), which are further subdivided into Th1, Th2, Th17 and Treg cells (Figure 3). The first three subtypes are effector T cells that act against intracellular pathogens (Th1), helminth parasites and other extracellular microbes (Th2) and extracellular bacteria and fungi (Th17) [189]. In addition, they actively participate in other non-infectious pathologies, such as allergies, autoimmune diseases or anti-tumor immune responses [189]. Conversely, Tregs are in charge of suppressing and controlling immune responses [190]. Although all these Th subtypes are present in the lamina propria, there is an enrichment of Foxp3^+^ Tregs and IL-17 producing Th17 cells in the lamina propria under homeostatic conditions [191]. In IBD, however, an imbalance of the homeostatic Th populations has been observed in the gut, with Th1-Th17/Th2 polarization of T CD4^+^ cell immune responses being involved in IBD pathophysiology [192].

T cells play a crucial role in immunosurveillance; CD8^+^ T cells can directly eliminate neoplastic cells, and CD4^+^ T cells collaborate with macrophages, NK cells and CD8^+^ T cells to mount anti-tumor immune responses [50,95,193]. In IBD-associated cancer, possible immunosurveillance functions of T cells in dysplastic lesions were postulated, associated with the expression of the co-stimulatory molecule CD80. Specifically, it was demonstrated that inhibition of CD80 signaling in vivo in a CAC mouse model significantly increased the frequency and size of high-grade dysplastic lesions, whereas restoration of CD80 expression decreased colonic dysplasia [194]. Another T-cell related immunological interaction relates to the CD30/CD30L axis. CD30L is expressed on T CD4^+^-activated cells [195]. Deletion of CD30L in an AOM/DSS-induced CAC mouse model promoted formation of an immunosuppressive tumor microenvironment, characterized by an increased percentage of PD-L1^+^ MDSCs and tolerogenic macrophages [196].

In addition, an earlier study comparing dysplastic lesions and cancers from patients with and without IBD showed that although IBD-associated dysplasia and cancer was associated with increased CD8^+^ T cell infiltration, these CD8^+^ T cells showed less expression of granzyme B and were thus less efficient in killing tumor cells [161] (Table 1). Furthermore, Yu et al. demonstrated that intratumoral CD8^+^ T cells in CAC mice display increased expression of the inhibitory markers PD-1 and CTLA-4, which may also indicate the presence of exhausted T cells [159]. This suppressive T-cell phenotype may contribute to tumor development, as effector cytokine production (IL-2, IFN-γ) is progressively lost in exhausted T cells [197], dampening effective immune responses against neoplastic cells. Similarly, upregulation of PD-1 expression in T CD8^+^ intraepithelial lymphocytes in mice subjected to AOM/DSS was identified [160] (Table 1), which was likely induced by repetitive cycles of inflammation. In humans, it was shown that patients with UC-associated dysplasia and cancer had increased expression of the PD-1 ligand PD-L1 on CD8^+^ T cells, as compared to sporadic CRC. PD-L1 overexpression correlated to chronic inflammation-induced DNA damage, and PD-L1 upregulation was mediated by inflammation-induced upregulation of IRF-1 [198] (Table 1). Whether this exhaustion phenotype on T cells also leads to decreased effector functions of T cells, and thus interferes with their immunosurveillance function, needs further investigation.

Lastly, Tregs have an important immunomodulatory role in limiting excessive inflammatory immune responses. They counteract the inflammatory response, but in the setting of cancer, expansion of the Treg pool may shape an immunosuppressive niche in which tumors can progress (Figure 2). In the context of IBD-associated cancer, studies in the AOM/DSS mouse model have shown that ablation of CD4^+^Foxp3^+^ Treg cells suppressed tumor growth, which was associated with increased numbers of CD8^+^IFN-γ^+^ Granzyme B-producing effector T cells [162,163] (Table 1). This demonstrates how immunosuppressive Tregs may be involved in the pathophysiology of IBD-associated cancer. Apart from immunosuppressive Tregs, IL17^+^Foxp3^+^CD4^+^ T cells, which can be induced by TGF-β, and IL-2 were identified as a functional proinflammatory Treg subpopulation present in the mucosal tissues of patients with active UC and in patients with UC-associated cancer, but not in non-inflammatory cancers, indicating that this subset of Tregs may also play a role in CAC pathogenesis [199].

Another interesting Treg subset in the setting of intestinal inflammation-induced cancer is the Foxp3^+^RORγt^+^ T cell. RORγt is a transcription factor other than Foxp3 which plays a role in Treg and Th17 differentiation. Studies have shown that RORγt^+^Treg cells promote inflammation and tumorigenesis by production of IL-17 [200], and they are observed in IBD and IBD-associated dysplastic lesions [201,202]. The tumor-promoting role of these cells has been demonstrated by Treg-specific deletion of RORγt in CAC mice, which resulted in decreased tumor incidence and decreased expression of ki67 and STAT3 in dysplastic lesions [202]. These data highlight the potential role for this Treg subset in IBD-associated tumorigenesis.

In summary, T cells exert a crucial function in killing neoplastic cells, but chronic inflammation as observed in IBD may induce an immunosuppressive environment with T-cell exhaustion and excessive Treg cell recruitment, which prevents adequate immunosurveillance and promotes evolution from non-dysplastic mucosa to dysplasia and/or carcinoma. In addition, pro-inflammatory Treg subpopulations such as IL-17-producing Tregs and Foxp3^+^RORγt^+^ Tregs may contribute to cancer development in IBD. It will be interesting to unravel the role of these T cell subsets in IBD-associated dysplasia and cancer in humans.

#### 4.2.5. Innate Lymphoid Cells

Innate lymphoid cells (ILCs) are a relatively recently discovered group of innate immune cells with diverse and important functions. They are generally classified in three types, with type 1 ILCs including NK cells. A large body of evidence demonstrates their role in IBD pathophysiology [203]. In line with this, patients with Crohn’s disease have increased numbers of NK cells in the small intestinal epithelial compartment [204] and colonic lamina propria [205].

Apart from their role in inflammation, NK cells have important anti-tumorigenic functions and play a pivotal role in clearing cancer cells [206,207]. In IBD-associated cancer, it has been proposed that NK cells participate in anti-tumor immunity through IL-15 stimulation produced by CD11c^+^ DCs [208]. Specifically, AOM/DSS-treated mice that deleted IL-15 showed reduced survival and higher tumor incidence. Conversely, reconstitution of IL-15 expression selectively in CD11c^+^ DCs restored NK cells and CD8^+^ T-cell compartments, with a subsequent reduction in tumor burden.

In the setting of chronic inflammation, NK cell immunosurveillance capacity may be compromised. In a recent study that addressed metabolic and functional profile of blood circulating NK cells in IBD patients, it was demonstrated that these NK cells produce pro-inflammatory cytokines such as IL-17A and TNF-α ex vivo, but they show limited killing capacity and defective mitochondrial activity [164] (Table 1). It could be hypothesized that such defects in NK cell killing capacity as observed in IBD patients might contribute to decreased immunosurveillance.

Additionally, type 3 ILCs (ILC3s), the innate counterparts of Th17 cells from the adaptive immune system, may be involved in IBD-associated cancer formation. ILC3s secrete cytokines including IL-23, IL-17 and IL-22, all of which have pro-tumorigenic effects [209,210,211]. Accumulation of IL-17^+^ IL-22^+^ ILC3s was identified in the colonic mucosa from CAC mice. The authors demonstrated the contribution of these cells to tumorigenesis in the context of UC, since its depletion blocks the development of invasive cancer lesions from dysplastic precursors. Further mechanistic analyses showed that IL-22 produced by colonic ILC3s in CAC mice acts on IECs to induce STAT3 phosphorylation [212]. Given the already known pro-cancer effect of STAT3 immune signaling [213], this study demonstrates an active role of ILCs in dampening anti-tumoral immune responses.

Thus, even though some ILCs, such as NK cells, are vital for mounting adequate anti-tumor immune responses, other ILCs including ILC3s may have a pro-tumorigenic effects via secretion of pro-inflammatory cytokines.

## 5. Conclusions

Long-standing chronic inflammation as observed in the intestinal mucosa of IBD patients increases the risk of CRC. The genetic and molecular changes in IECs of IBD-associated cancers are well characterized, and the role of immune cell-derived pro-inflammatory mediators has been studied extensively, mostly using mouse models.

Less studied potential mechanisms of CAC are the anti-inflammatory/immunosuppressive pathways that emerge during long lasting inflammation, which may create a niche for tumors to grow. It is already well-known that cancers can promote an immunosuppressive microenvironment favoring its own growth and progression. In recent years, this phenomenon has gained more attention, mainly due to the noticeable anti-tumor effects of immune checkpoint inhibitors, which revert to this immunosuppressive state and activate the immune system. Here, we provide an overview of different immune cells that are recruited into the inflamed mucosa of IBD patients and describe how these immune cells impact cancer development, either by anti- or pro-tumorigenic effects. For example, increased expression of T-cell inhibitory receptors such as PD-1 and CTLA-4 have been observed in intratumoral CD4^+^ and CD8^+^ T cells lesions in mouse models for inflammation-driven colorectal cancer, indicating that T-cell suppression may be involved in the etiology of IBD-associated cancer. Although there are only limited data in humans, the chronic inflammation-induced immunosuppressive environment may thus be another mechanism driving colitis-associated cancer, not only by induction of dysfunctional, exhausted T cells, but also by the recruitment of Tregs, MDSCs and other suppressor cells into the inflamed intestinal mucosa.

It is important to understand the immunosuppressive mechanisms that evolve during chronic inflammation and may be involved in cancer development by allowing cancers to evade anti-tumor immune responses. This is especially the case for patients at high risk for developing IBD-associated dysplasia and cancer, such as patients with IBD and PSC. Additionally, better understanding of the mechanisms involved may dictate the choice of immunosuppressive drugs used in these patients.

## Figures and Tables

**Figure 1 ijms-22-12739-f001:**
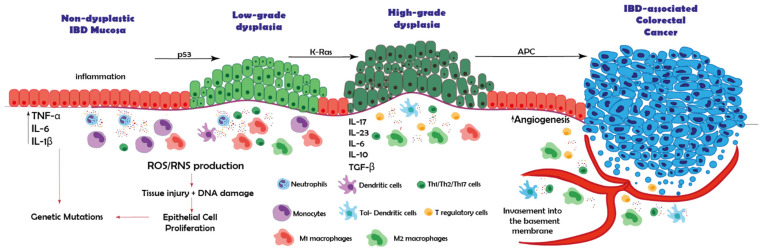
Schematic representation of the contribution of the immune system in the dysplasia to carcinoma sequence in IBD-associated cancer. Chronic inflammation promotes the recruitment of immune cells into the intestinal mucosa of inflammatory bowel disease (IBD) patients. These cells will secrete pro-inflammatory cytokines such as tumor necrosis factor alpha (TNF-α), interleukin (IL) -6 and IL-1β. In addition, these tissue-infiltrating cells produce oxidative compounds including reactive oxygen species and reactive nitrogen species, generating tissue injury and DNA damage, promoting excessive epithelial cell proliferation and favoring genomic aberrations and genetic mutations. TP53 mutations result in low-grade dysplastic mucosa, after which mutations in KRAS are considered to be involved in progression from low-grade dysplasia to high-grade dysplasia. Finally, mutations in the APC gene result in cancer. Chronic inflammation also induces immunosuppressive mechanisms that may be involved in cancer progression, such as recruitment of M2 macrophages, T regulatory cells (Tregs) or T lymphocytes expressing inhibitory markers; programmed cell death protein 1 (PD-1), cytotoxic T-lymphocyte-associated protein 4 (CTLA-4). In addition, there is an increase in angiogenic factors and anti-inflammatory cytokines; IL-10 and transforming growth factor beta (TGF-β) which favor carcinoma development. (Tol-Dendritic cells: Tolerogenic dendritic cells, ROS: reactive oxygen species, RNS: reactive nitrogen species).

**Figure 2 ijms-22-12739-f002:**
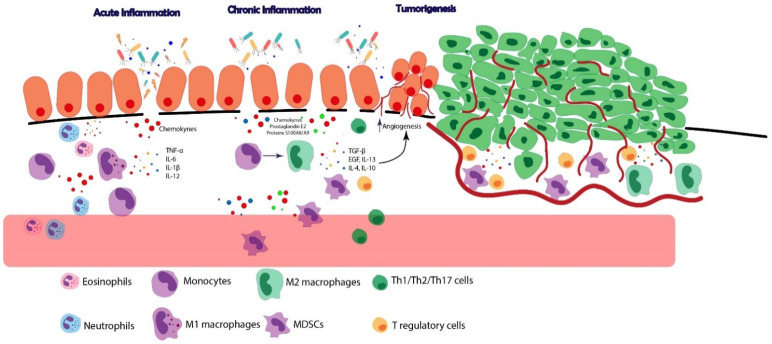
Schematic representation of chronic inflammation-associated carcinogenesis. Tissue injury, microbial infections or autoimmune diseases trigger acute inflammatory responses, characterized by an abundance of neutrophils, eosinophils and monocytes, which are recruited into the tissue by a variety of chemokines. Together with M1 macrophages these immune cells produce pro-inflammatory cytokines (TNF-α, IL-6, IL-1β, IL-12) leading to tissue damage. When inflammation becomes chronic, adaptation occurs with a shift towards increased numbers of macrophages and lymphocytes among tissue-infiltrating cells, and alteration of the inflammatory milieu with other inflammatory compounds such as prostaglandin E2 and S100A8/9 proteins. In addition, immune-regulatory and anti-inflammatory mechanisms emerge, which may promote carcinoma development, such as increased M2 macrophage differentiation from monocytes, and recruitment of immunosuppressive cells; myeloid-derived suppressor cells (MDSCs) and Tregs. M2 macrophages produce anti-inflammatory cytokines (TGF-β, EGF, IL-13, IL-10) which can, for example, promote angiogenesis and fibrosis, favoring tumor growth. Consequently, the tumor itself creates an immunosuppressive milieu, characterized by elevated levels of M2 macrophages, MDSCs and Tregs. (EGF: epidermal growth factor).

**Figure 3 ijms-22-12739-f003:**
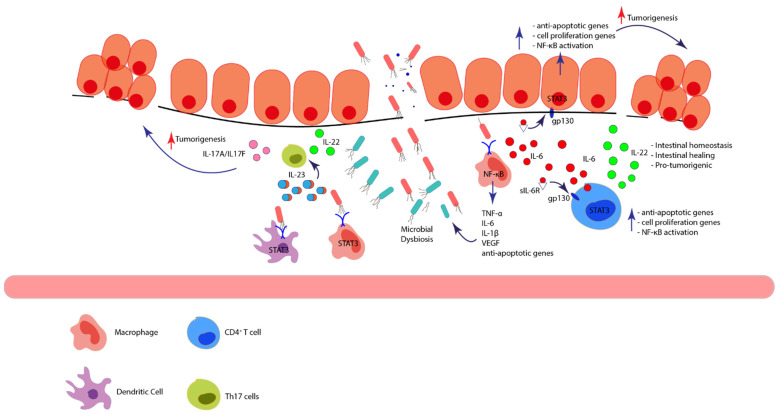
Pro-inflammatory immune signaling pathways in IBD and their contribution to IBD-associated cancer. Disruption of the intestinal epithelial layer causes the entry of luminal microorganisms. Pathogen-specific structures are recognized in the intestinal mucosa via Toll-like receptors by innate immune cells, activating the transcription factor NF-κB and promoting TNF-α, IL-6 and IL-1β production. NF-κB also stimulates transcription of vascular endothelial growth factor and a number of anti-apoptotic genes, enhancing angiogenesis, cell survival and proliferation. IL-6 has a potent pro-tumorigenic effect, mainly via STAT3 signaling. It binds to soluble or membrane-bound receptors which interact with gp130 either in intestinal epithelial cells or immune cells, triggering the activation of signaling molecules and transcription factors, including STAT3. STAT3 regulates the transcription of genes related to cell proliferation and survival and stimulates activation of NF-κB, creating a positive feedback loop. It also stimulates production of IL-22 by CD4^+^ T cells, which is important in intestinal homeostasis but potentially also facilitates carcinogenesis. Microbial recognition by antigen presenting cells also stimulates production of IL-23 via STAT3 signaling, leading to Th17 polarization with the subsequent production if IL-17A, IL-17F and IL-22, cytokines associated with carcinogenesis. (TLRs: toll-like receptor, VEGF: vascular endothelial growth factor).

**Table 1 ijms-22-12739-t001:** Possible anti-inflammatory and immunosuppressive mechanisms that may dampen immunosurveillance during IBD-associated carcinogenesis.

Immune Cell	Mechanism
**Macrophages**	− Accumulation of CD163^+^ M2 macrophages that activate oncogenic pathways via IL-13 and CCL17 in UC and CAC patients [153].
	− Glycan-induced expression on colonic epithelial cells by CD163^+^ M2 macrophages that inhibit NK and DC function in UC and CAC patients [153].
	− Inhibition of T-cell responses via iNOS^+^ macrophages [154]. − TGF-β production by PPAR-γ expressing macrophages [155].
**Dendritic** **Cells (DCs)**	− Notch-2- DCs with less migration to MLNs and antigen presentation capacity to CD8^+^ T cells [156].
	− Plasmacytoid dendritic cells (pDCs) induce recruitment of MDSCs into intestinal mucosa in a CAC mouse model [157].
**Myeloid-derived suppressor cells (MDSCs)**	− CXCR2-expressing MDSCs in mice colonic mucosa inhibit CD8^+^ T effector functions [158].
**T cells**	− Increased expression of inhibitory markers (PD-1, CTLA-4, PD-L1) in CD8^+^ T cells in IBD-associated cancer tumors and intestinal epithelium from CAC mice [159,160].
	− Decreased granzyme B expression in CD8^+^ T cells in IBD-associated cancer patients [161].
	− Tregs suppress CD8^+^IFN-γ^+^ T cells producers of granzyme B in a CAC mouse model [162,163].
**Natural Killer (NK) cells**	− Limited killing capacity and mitochondrial activity of circulating NK cells from IBD patients [164].

## Data Availability

Not applicable.

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
