# Peer review of "The Role of the Immune System in IBD-Associated Colorectal Cancer: From Pro to Anti-Tumorigenic Mechanisms"

_ijms, 2021, doi:10.3390/ijms222312739_

Round 1
Reviewer 1 Report
This review manuscript summarizes the role of the immune system in IBD-associated ( Inflammatory Bowel Disease ) cancer with a particular focus on the role of immunosuppressive mechanisms that may be involved in the development of CRC in IBD patient.
The manuscript seems well written but in my opinion should be improved in several aspects.(please see the comments).
Major revision:
1) The Authours should use include recent paper and sicuss the results like (https://doi.org/10.193/ecco-jcc/jjab102)
2) The Authours should improve the section 4 (4. Immune cells in IBD and its contribution to IBD-associated cancer) with revelant articles on the immune cells studies.
3) The Authours should improve with the iformation of therapeutic agents used and studies in this disease.
Author Response
1) The Authours should use include recent paper and sicuss the results like (https://doi.org/10.193/ecco-jcc/jjab102)
Thank you for bringing this interesting review to our attention. We have referenced it in the manuscript (lines 45 to 48).
2) The Authours should improve the section 4 (4. Immune cells in IBD and its contribution to IBD-associated cancer) with revelant articles on the immune cells studies.
We have again studied the literature for additional relevant articles concerning the immune cells. We believe we have been quite extensive but as requested we have added more studies on how immune cell subsets and their products contribute to the pathophysiology of IBD-associated dysplasia and cancer. Please see the red-coloured parts in section 4.2 in the revised manuscript.
3) The Authours should improve with the iformation of therapeutic agents used and studies in this disease.
We have added a paragraph about conventional therapies used in IBD and its impact on IBD-associated cancer in section 2.
Reviewer 2 Report
The review article is of high interest and the sections look balanced, scoping an important topic for IBD associated cancer research, such as pro- and anti-inflammatory priming. Beyond summaries of important scientific developments and ideas, authors tried to identify cell types and discuss how secretory mediators may contribute in neoplastic transformation. This manuscript is well conceptualized and written, however authors still need to address some points to make this manuscript more interesting.
- In the Introduction section authors should also provide some background on Chron’s disease in the progression of Cancer.
- A separate section is required to highlight different transcription factors associated with IBD associated cancer development and any reported therapeutic importance.
- Anti-TNF therapy is a most effective IBD treatment in the current scenario, authors should discuss how this pro-inflammatory molecule will modulate cancer progression.
- Figure 3 legends: need better representation in the text with mention at the specific points.
- Section 3: Need to discuss JAK-STAT pathway here.
- Chronic inflammation further leads to microbial dysbiosis and hyperinflammatory stage at the mucosal site, how this condition contributes to cancer progression or suppression?
- Section 4.4: Authors should clearly state their opinion whether IBD condition leads to suppression of antigen presentation mechanism in cancer cell to escape from T cell recognition?
Author Response
- In the Introduction section authors should also provide some background on Chron’s disease in the progression of Cancer.
We have introduced the requested information in section 1 and 2 of the revised manuscript (lines 40-41; lines 91-96).
- A separate section is required to highlight different transcription factors associated with IBD associated cancer development and any reported therapeutic importance.
Although it may be somewhat beyond the scope of our review, we definitely acknowledge the importance of these factors in IBD-associated cancer. We have added a section about different immune signaling pathways and its implications in IBD and IBD-associated cancer.
- Anti-TNF therapy is a most effective IBD treatment in the current scenario, authors should discuss how this pro-inflammatory molecule will modulate cancer progression.
We agree with the importance of anti-TNF therapy in IBD and the possibility that it modulates cancer progression. We have discussed this issue in section 4.1.
- Figure 3 legends: need better representation in the text with mention at the specific points.
We understand the reviewer’s comment. We have included a new Figure 3, and omitted the previousFigure 3. In addition, the mechanisms of immunosuppression are depicted in a table format (see table 1).
- Section 3: Need to discuss JAK-STAT pathway here.
We have discussed this pathway in section 4.1 (particularly STAT3 downstream of JAK).
- Chronic inflammation further leads to microbial dysbiosis and hyperinflammatory stage at the mucosal site, how this condition contributes to cancer progression or suppression?
Thank you for raising this interesting point. Although we certainly agree that this is of interest, we believe this is a subject that is beyond the scope of the current review. The role of microbial dysbiosis in cancer and IBD-associated cancer progression has been excellently discussed in:
Sheflin AM, Whitney AK, Weir TL. Cancer-promoting effects of microbial dysbiosis. Curr Oncol Rep. 2014 Oct;16(10):406.
Sobhani I, Amiot A, Le Baleur Y, Levy M, Auriault ML, Van Nhieu JT, Delchier JC. Microbial dysbiosis and colon carcinogenesis: could colon cancer be considered a bacteria-related disease? Therap Adv Gastroenterol. 2013 May;6(3):215-29.
Dalal N, Jalandra R, Bayal N, Yadav AK, Harshulika, Sharma M, Makharia GK, Kumar P, Singh R, Solanki PR, Kumar A. Gut microbiota-derived metabolites in CRC progression and causation. J Cancer Res Clin Oncol. 2021 Nov;147(11):3141-3155.
Popov J, Caputi V, Nandeesha N, Rodriguez DA, Pai N. Microbiota-Immune Interactions in Ulcerative Colitis and Colitis Associated Cancer and Emerging Microbiota-Based Therapies. Int J Mol Sci. 2021 Oct 21;22(21):11365.
7. Section 4.4: Authors should clearly state their opinion whether IBD condition leads to suppression of antigen presentation mechanism in cancer cell to escape from T cell recognition?
We have stated our view and opinion about chronic inflammation and immunosuppression in the conclusion of our review. The main message however remains that we believe that this is a quite novel and understudied concept that needs to be addressed in future studies.
For example, although it is well known that tumors can evade destruction by reducing immunogenicity by for example reducing capacity to upregulate MHCI, mainly genetic mechanisms such as gene inactivation have been implied for this defect in MHCI upregulation. However, other studies have shown that, apart from basal reduction in MHCI, reduction in pro-inflammatory cytokines including IFN-γ could cause decrease expression of MHCI in neoplastic cells. Also, the adaptive immune response could edit tumor antigen expression, as it has been shown that for example CD4+ T cells were able to directly down-regulate tumor antigen expression. Such mechanisms can certainly play a role in chronic inflamed environment characteristic of IBD and may also affect antigen presentation in dysplastic and neoplastic cells, yet this still needs further investigation. In addition, the presence of immunosuppressive cells in chronically inflamed mucosa may create a tolerogenic environment which hampers antigen recognition and consequent pro-inflammatory responses in general.
Round 2
Reviewer 1 Report
No other comments.
Reviewer 2 Report
Authors have addressed all the comments, I have no further comments.